# The Effects of Electrolytic Multivitamins and Neomycin on Antioxidant Capacity and Intestinal Damage in Transported Lambs

**DOI:** 10.3390/ani14060824

**Published:** 2024-03-07

**Authors:** Cui Xia, Chunhui Duan, Conghui Chen, Xinyu Yang, Yingjie Zhang, Yueqin Liu, Yuzhong Ma

**Affiliations:** 1College of Animal Science and Technology, Agricultural University of Hebei, Baoding 071001, China; xiacui950114@163.com (C.X.); duanchh211@126.com (C.D.); cchpluto@163.com (C.C.); zhangyingjie66@126.com (Y.Z.); 2College of Veterinary Medicine, Agricultural University of Hebei, Baoding 071001, China; b18803308783@163.com

**Keywords:** transport stress, lamb, antioxidant, intestinal injury, electrolytic multivitamin, neomycin

## Abstract

**Simple Summary:**

Transportation stress can lead to a decrease in immune function and induce various oxidative stresses, which affect health and productive performance. In this experiment, lambs were fed diets containing electrolytic multivitamins and neomycin. The activities of the antioxidant enzymes SOD, GSH-Px, and CAT, and the levels of MDA and T-AOC in sera, were examined. At the same time, the activities of antioxidant enzymes SOD and GSH-Px, and the levels of MDA and T-AOC in the jejunum and colon, as well as the mRNA expressions of SOD, CAT, Nrf2, HO-1, Keap1, IL-1, IL-2, IL-12, Bax, Bcl-2, and Caspase3 in the jejunum and colon, were measured. In addition, the contents of IgA, IgG, IgM, and sIgA in the jejunum and colon were examined. It was found that road transport can decrease the antioxidant capacity and contents of immunoglobulin and increase the expression levels of inflammatory factors and apoptosis in the jejunum and colon of lambs. Electrolytic multivitamins had a better effect on improving antioxidant activity, inhibiting the expression of inflammatory factors in lambs, and potentially reducing the expression levels of apoptotic factors and oxidative damage to the jejunum and colon.

**Abstract:**

Transport stress can cause damage to animals. In this experiment, 60 four-month-old lambs were randomly divided into three groups: CG (basal diet), EG (basal diet + 375 mg/d/lamb electrolytic multivitamin), and NG (basal diet + 200 mg/d/lamb neomycin). The results were as follows: during road transport, in all groups, the levels of SOD, T-AOC, and GSP-Px, and mRNA expressions of CAT, SOD, Nrf2, HO-1, and Bcl-2 in the jejunum and colon decreased (*p* < 0.01). However, mRNA expressions of Keap1, IL-1β, IL-2, IL-12, Bax, and Caspase3 in the jejunum and colon and the level of MDA increased (*p* < 0.01). The concentrations of IgA, IgG, and sIgA in the jejunum and colon also decreased (*p* < 0.01). In the EG and NG, the levels of SOD (*p* < 0.05) and T-AOC (*p* < 0.01) increased, and the level of MDA decreased (*p* < 0.01). However, in the jejunum, the levels of SOD and T-AOC, the concentrations of IgA and IgG, and mRNA expression of Bcl-2 increased (*p* < 0.05). mRNA expressions of IL-1, IL-2, and Caspase 3 (*p* < 0.05), and mRNA expression of IL-12 (*p* < 0.01) decreased. In the colon, SOD activity and the concentration of sIgA increased (*p* < 0.01). The level of MDA and mRNA expressions of IL-2 and Caspase 3 also decreased (*p* < 0.05). In the jejunum and colon, mRNA expression of SOD (*p* < 0.05) and mRNA expression of Nrf2 increased (*p* < 0.01). mRNA expression of Keap1 (*p* < 0.05) and Bax (*p* < 0.01) decreased. In summary, road transport can cause a decrease in antioxidant activity and immunity of lambs and an increase in oxidative damage. Electrolytic multivitamins and neomycin can improve immune function and potentially reduce oxidative damage to the jejunum and colon.

## 1. Introduction

Transportation is an essential link in husbandry production; it is also one of the most common stressors in animals. Research shows that Japanese black cattle experienced transport stress after they were transported for 6 h [1]. Various transport stressors can lead to various infectious diseases [2] and even death [3] in animals, which brings economic losses and restricts the development of husbandry. Under stress, the body will produce more free radicals, which can lead to oxidative damage in animals. The intestine is an important site for digestion, absorption, and nutrient metabolism; the stability of intestinal function is crucial for the body. However, the intestine can easily become a target organ for stress. Large amounts of reactive oxygen species (ROS) accumulate in intestinal epithelial cells during stress, which disrupts the antioxidant system and leads to oxidative stress. Wei et al. [4] showed that intestinal oxidative stress occurred, the antioxidant system was destroyed, intestinal barrier function was damaged, and the incidence rate increased after pigs experienced transport stress. In addition, oxidative stress can regulate the expression of apoptosis genes, mediate apoptosis, and damage the intestinal barrier of animals [5,6]. It increases the expression of inflammatory factors, inhibits immune function [7], and affects normal metabolism.

At present, only a limited number of papers investigating the effects of transport stress on antioxidant capacity and intestinal damage in lambs are available. Research has shown that a certain amount of vitamins could effectively alleviate transport stress [8,9]. Transport stress causes infectious disease and increases the incidence rate. Antibiotics are the first choice for the treatment of bacterial diseases [10]. A lack of research is observed in the literature regarding the effects and mechanisms of electrolytic multivitamins and neomycin on antioxidant capacity and intestinal damage in transported lambs. Regarding antioxidant capacity and intestinal damage, research is mainly focused on cattle [11] and pigs [4]. Our previous research found that electrolytic multivitamins and neomycin could potentially affect the intestinal barrier function of lambs [12]. The purpose of this study is to evaluate the effects of electrolytic multivitamins and neomycin on antioxidant capacity and intestinal damage in transported lambs and explore the possible mechanisms. It provides new ideas for improving antioxidant capacity and reducing intestinal damage in transported lambs.

## 2. Materials and Methods

### 2.1. Experimental Design and Feeding Management

A total of 60 lambs (average live weight, 18.99 ± 0.54 kg, 4 months old) were randomly divided into three equal groups. The control group (CG) was fed a basal diet, and the composition of the basal diet is shown in Table 1. This experiment was based on previous reports [13,14]; preliminary experiments were conducted in the early stage. According to the preliminary experimental results, the best time for adding electrolytic multivitamins was from 2 days before transportation to 7 days after transportation, and the best time for adding neomycin was from 0 to 7 days after transportation. The electrolytic multivitamin group (EG) was fed a basal diet with 375 mg/d of electrolytic multivitamin per lamb (Anhui Lingmu Biotechnology Co., Ltd., Fuyang, China); the main components and contents of electrolytic multivitamins are shown in Table 2. The neomycin group (NG) was fed a basal diet of 200 mg/d of neomycin per lamb (Henan Weilong Veterinary Medicine Co., Ltd., Kaifeng, China). The experimental lambs were transported for 8 h from 09:00 to 17:00 on 28 July. The day of transportation was recorded as the first day of the experiment (day 0). Five lambs in each group were randomly selected on days −2, 0, 7, and 14, then 5 mL of jugular venous blood was collected for separating sera at 3500 r/min for 5 min; it was placed in sterile tubes and stored at −20 °C. After blood collection, lambs were anesthetized with a dose of 0.2 mg/kg of xylazine (Hebei Mingeng Biotechnology Co., Ltd., Shijiazhuang, China) on day −2, 0, 7, and 14, and exsanguinated. The middle of the jejunal and colonic tissues were cut and collected, then rinsed the contents with physiological saline. The tissues were placed in sterile tubes and frozen in liquid nitrogen. The approval number for this experiment was 2021023.

The lambs were bought from Zhangjiakou Lanhai Livestock Breeding Co., Ltd. (40°37′ N 115°03′ E and ~1300–1600 m above sea level, Zhangjiakou, China). The vehicle was a truck with a density of 1.7 m^2^/lamb. The speed of the vehicle was 30–75 km/h. On the day of transportation, the outdoor temperature was 18–29 °C, and the relative humidity was 64%. 

The pens and breeding tools were disinfected thoroughly. Each group of lambs was raised separately from 1 July to 30 July. Before feeding, the electrolytic multivitamins and neomycin were mixed thoroughly with a small amount of basal diet to ensure drugs were consumed completely by each lamb so all lambs could eat and drink freely. Lambs were fasted for 10 h before transportation, and fresh water was freely provided until transportation. Only water was given within 12 h after transportation, and a basal diet was provided after 12 h. The basal diet was included by the sheep farm; lambs were fed twice a day at 7:00 and 17:00 to ensure that they all had leftover feed in their feed tanks every day.

### 2.2. Determination of Antioxidant Capacity in Sera and Intestines

The indicators of the antioxidant capacity, such as superoxide dismutase (SOD), glutathione peroxidase (GSH-Px), catalase (CAT), malondialdehyde (MDA), and total antioxidant capacity (T-AOC) were measured according to the instructions of the detection kit (Nanjing Jiancheng Bioengineering Institute, Nanjing, China).

### 2.3. Determination of Intestinal Immunoglobulins

The concentrations of immunoglobulin A (IgA), immunoglobulin G (IgG), secretory immunoglobulin A (sIgA), and immunoglobulin M (IgM) in the jejunum and colon were measured by the enzyme-linked immunosorbent assay (ELISA) detection kit (Nanjing Jiancheng Bioengineering Institute, Nanjing, China).

### 2.4. mRNA Expressions Analysis by Quantitative Real-Time PCR

Total RNA was isolated from the jejunum and colon (50 mg) with Trizol reagent (TransGen Biotech, Beijing, China). The concentration of total RNA in the samples was evaluated using a spectrophotometer (NanoDrop-2000, Thermo Fisher Scientific, Waltham, MA, USA) at 260 nm and 280 nm, respectively. Ratios of absorption (260:280 nm) between 1.8 and 2.0 for all samples were accepted as “pure” for RNA. Reverse transcription of RNA into cDNA by the Reverse Transcription Kit was performed according to the instructions (Transgen Biotech, Beijing, China). Primer sequences for IL-1β, IL-2, IL-12, CAT, SOD, Nrf2, Keap1, HO-1, Bax, Bcl 2, and Caspase-3 in the jejunum and colon were designed using the GenBank database from the National Center for Biotechnology Information and software of primer 5 design. RT-qPCR was performed with SYBR Green Premix Es Taq (Takara, Beijing, China) using the Step One Plus real-time PCR system (Applied Biosystems (Thermo-Fisher Scientific, Shanghai, China)) on 96-well plates with 20 µL total reaction volume, which included 10 µL of SYBR Green Premix, 1 µL of cDNA, 0.4 µL of forward primers, 0.4 µL of reverse primers, and 8.2 μL of double-distilled water. Each reaction was run in duplicate. The PCR cycling protocol included one cycle of pre-incubation at 95 °C for 2 min, 40 cycles of denaturation at 95 °C for 5 s, and annealing at 60 °C for 30 s; each cycle increased by 0.5 °C at 65–95 °C. β-actin was used as a reference, and the relative expression of the target gene to β-actin was determined using the 2^−ΔΔCt^ method, as described by Livak [15]. Primers for RT-qPCR were synthesized by the Beijing Genomics Institution (BGI Genomics Co., Ltd., Beijing, China; Table 3).

### 2.5. Statistical Analysis

All data were analyzed using single-factor ANOVA in SPSS 21.0 statistical software [16] to determine differences in indicators when different treatments were administered at the same time versus the same treatment at different times. Duncan’s method was used for multiple comparisons. The General Linear Model (GLM) was used to analyze the effects of treatments and time interaction. Effects *p* ≤ 0.05 were considered different. Values were expressed as “Mean ± Standard Error” (M ± SEM).

## 3. Results

### 3.1. Effects of Different Treatments on the Antioxidant Capacity of Transported Lambs

#### 3.1.1. Effects of Different Treatments on the Antioxidant Capacity in Sera of Transported Lambs

As shown in Table 4, after lambs were transported, the levels of SOD and T-AOC in the CG and NG decreased (*p* < 0.01), while the level of MDA increased (*p* < 0.01). On day 0, compared with the CG, the SOD activity in the EG (*p* < 0.05), as well as the activity of T-AOC in the EG, increased (*p* < 0.01). On day 14, the level of MDA in the EG was lower than that in the CG (*p* < 0.05), and the level of T-AOC in the EG was higher than those in the CG and NG (*p* < 0.01). There was no difference in the levels of MDA in the CG and NG on day 7, and SOD and T-AOC in the NG on day 14 compared with those on day −2d (*p* > 0.05).

#### 3.1.2. Effects of Different Treatments on the Antioxidant Capacity of Jejunum and Colon in Transported Lambs

As shown in Table 5, by transportation, the levels of GSH-Px and T-AOC in the jejunum and colon of the EG and NG, as well as the SOD activity in the jejunum and colon of each group, decreased (*p* < 0.01). The level of MDA in the jejunum of each group, as well as those in the colon of the CG and NG, increased (*p* < 0.01). On day 0, the level of MDA in the jejunum of the EG was lower than that of the CG (*p* < 0.01). In the colon, the SOD activity of the EG was higher than those of the CG and NG (*p* < 0.01), and the MDA of the EG was lower than those of the CG and NG (*p* < 0.01). On day 7, in the jejunum, the SOD activity of the EG was higher than that of the CG (*p* < 0.05), and the level of MDA was lower than that of the CG (*p* < 0.01). The SOD activity in the colon of the EG and NG was higher than that of the CG (*p* < 0.01). On day 14, the level of T-AOC in the jejunum of the EG and NG was higher than that of the CG (*p* < 0.05), and the SOD activity in the colon of the EG and NG was higher than that of the CG (*p* < 0.01). The levels of MDA in the jejunum of the EG, T-AOC in the jejunum of the NG, and MDA in the colon of the CG and NG on day 7 had no significant changes compared with those on day −2 (*p* > 0.05). In the jejunum, there was no difference in GSH-Px activity of the CG and NG, SOD activity of the EG, and MDA in the NG on day 14 compared with those on day −2 (*p* > 0.05). In the colon, the GSH-Px activity of the NG and SOD of the EG and NG on day 14 had no significant changes compared with those on day −2 (*p* > 0.05).

#### 3.1.3. Effects of Different Treatments on the mRNA Expression of Antioxidant Factors in the Jejunum and Colon of Transported Lambs

As shown in Table 6, by transportation, the mRNA expression of CAT in the jejunum and colon of each group, SOD in the jejunum of each group, and SOD in the colon of the CG and NG decreased (*p* < 0.01). On day 7, the mRNA expression of SOD in the jejunum of the EG was higher than that of the CG (*p* < 0.01). On day 14, the mRNA expressions of SOD in the jejunum and colon of the EG were higher than those of the CG (*p* < 0.05). On day 7, there was no difference in the mRNA expression of SOD in the jejunum of the EG compared with that on day −2 (*p* > 0.05). On day 14, the mRNA expressions of CAT in the jejunum and colon of the EG, as well as the mRNA expression of SOD in the jejunum of the NG, had no significant changes compared with those on day −2 (*p* > 0.05).

#### 3.1.4. Effects of Different Treatments on mRNA Expression of Key Factors in the Nrf2 Pathway in the Jejunum and Colon of Transported Lambs

As shown in Table 7, by transportation, the mRNA expressions of Nrf2 and HO-1 in the jejunum and colon of each group decreased (*p* < 0.01). The mRNA expressions of Keap1 in the jejunum of the CG and NG, as well as the mRNA expression of Keap1 in the colon of each group, increased (*p* < 0.01). On day 0, the mRNA expression of Nrf2 in the jejunum of the EG was higher than those of the CG and NG (*p* < 0.01), while Keap1 in the jejunum of the EG was lower than that of the CG (*p* < 0.05). On day 7, the mRNA expression of Keap1 in the colon of the EG was lower than that of the CG (*p* < 0.05). On day 14, the mRNA expressions of Nrf2 in the jejunum and colon of the EG were higher than that of the CG (*p* < 0.01).

On day 7, the mRNA expressions of Keap1 in the jejunum of the CG and NG, as well as the mRNA expression of Keap1 in the colon of the EG, had no significant changes compared with those on day −2 (*p* > 0.05). On day 14, there was no difference in the mRNA expressions of Nrf2 in the jejunum and colon of the EG and NG compared with those on day −2 (*p* > 0.05). The mRNA expressions of HO-1 in the jejunum and colon of the EG had no significant change compared with that on day −2 (*p* > 0.05). There was no significant difference in the mRNA expressions of Keap1 in the colon of the CG and NG compared with that on day −2 (*p* > 0.05).

### 3.2. Effects of Different Treatments on the Intestinal Immune Function of Transported Lambs

#### 3.2.1. Effects of Different Treatments on Intestinal Immunoglobulin of Transported Lambs

As shown in Table 8, by transportation, the concentrations of IgA, IgG, and sIgA in the jejunum and colon of the CG and NG, as well as the concentration of sIgA in the jejunum of the EG decreased (*p* < 0.01). The concentration of sIgA in the jejunum of the EG on day 0, as well as the concentration of sIgA in the colon of the EG, was higher than those in the CG and NG on days 0, 7, and 14 (*p* < 0.01). The concentrations of IgG in the jejunum of the EG on day 7 and IgA in the jejunum of the EG on day 14 were higher than those of the CG (*p* < 0.05). The concentrations of sIgA in the jejunum of the EG and NG were higher than that of the CG on days 7 and 14 (*p* < 0.01).

On day 7, there was no significant difference in the concentrations of IgA and IgG in the jejunum and colon of the NG and sIgA in the EG compared with those on day −2 (*p* > 0.05). On day 14, the concentration of IgG in the jejunum and colon of the CG had no significant change compared with those on day −2 (*p* > 0.05). There was no significant difference in the concentration of sIgA in the jejunum of the NG and IgA in the colon of the CG compared with those on day −2 (*p* > 0.05).

#### 3.2.2. Effects of Different Treatments on mRNA Expression of Inflammatory Factors in Jejunum and Colon of Transported Lambs

As shown in Table 9, by transportation, the mRNA expressions of IL-1β and IL-2 in the jejunum of each group, as well as the mRNA expressions of IL-12 in the jejunum and IL-1β, IL-2, and IL-12 in the colon of the CG and NG increased (*p* < 0.01). The mRNA expressions of IL-1β in the jejunum of the EG on day 14 and IL-2 in the colon of the EG on day 0 were lower than those of the CG and NG (*p* < 0.05). On day 14, the mRNA expressions of IL-2 in the jejunum of the EG and NG (*p* < 0.05), as well as the mRNA expression of IL-12 in the jejunum of the EG and NG was lower than those of the CG (*p* < 0.01).

On day 7, the mRNA expressions of IL-1β and IL-2 in the jejunum of the EG, IL-1β, and IL-12 in the colon of the CG and NG, and IL-2 in the colon of the NG had no significant changes compared with those on day −2 (*p* > 0.05). On day 14, there was no significant difference in the mRNA expressions of IL-1β, IL-2, and IL-12 in the jejunum of the NG, IL-12 in the jejunum of the EG, and IL-2 in the colon of the CG compared with those on day −2 (*p* > 0.05).

### 3.3. Effects of Different Treatments on Apoptosis Factors in the Jejunum and Colon of Transported Lambs

As shown in Table 10, by transportation, the mRNA expressions of Bax in the jejunum and colon and Caspase3 in the jejunum of each group increased (*p* < 0.01), while the mRNA expression of Bcl-2 decreased (*p* < 0.01). On day 0, the mRNA expressions of Bax in the jejunum and colon of the EG were lower than that of the CG (*p* < 0.01), and Caspase3 in the jejunum of the EG was lower than those of the CG and NG (*p* < 0.01). On day 7, the mRNA expressions of Bax in the jejunum and colon of the EG and Caspase3 in the colon of the EG were lower than those of the CG (*p* < 0.01). On day 14, the mRNA expression of Bcl-2 in the jejunum of the EG was higher than that of the CG (*p* < 0.05).

On day 7, the mRNA expressions of Bax in the jejunum and colon of the EG, Bcl-2 in the colon of each group, and Caspase3 in the colon of the NG had no significant changes compared with those on day −2 (*p* > 0.05). On day 14, in the jejunum, there was no difference in the mRNA expressions of Bax in the NG, Bcl-2 in the EG and NG, and Caspase3 in the NG compared with those on day −2 (*p* > 0.05). In the colon, the mRNA expressions of Bax of the CG and NG and Caspase3 of the CG had no significant changes compared with those on day −2 (*p* > 0.05).

## 4. Discussion

Transportation is an important link in the development of the sheep industry; during transportation, the metabolic level increases to resist the damage caused by stress, which leads to excessive production of ROS in cells and oxidative damage to the body. The degree of oxidative damage can be determined by measuring the activity of the antioxidant enzymes and the level of MDA [17]. Earlier research showed that the activities of GSH Px, T-SOD, CAT, and T-AOC reduced significantly, and the MDA levels increased significantly after animals experienced heat stress [18,19]. The concentration of MDA reflects the degree of oxidative damage, which is one of the important indicators that reflect oxidative stress [20]. The decrease in T-AOC level indicates an imbalance in the production and clearance of free radicals in the body. The SOD, CAT, and GSH-Px are the first line of defense against antioxidant stress. In this experiment, by road transport, the levels of GSH-Px, CAT, SOD, and T-AOC of lambs decreased significantly, while the level of MDA increased significantly. The results showed that the antioxidant capacity decreased significantly, and oxidative damage increased after lambs were transported; road transport may cause oxidative stress in lambs. The levels of the antioxidant enzymes and MDA in sera and intestines, and the expressions of SOD and CAT in intestines showed consistent trends; road transport may inhibit the activity of the antioxidant enzyme in lambs by downregulating their relative expressions, which accelerated lipid peroxidation reactions in cells and tissues. Road transport led to large amounts of free radicals produced; SOD, CAT, and GSH-Px were consumed greatly to eliminate free radicals, which accelerated lipid peroxidation in the body, so the levels of GSH-Px, CAT, SOD, and T-AOC decreased, and MDA increased, potentially damaging intestinal structure and function [21]. Other research shows that moderate vitamins can inhibit the damage of oxidative stress and maintain the redox balance of the body [22,23]. In this experiment, using electrolytic multivitamins and neomycin treatments, the levels of SOD and T-AOC increased after lambs were transported, while the level of MDA decreased. The results showed that electrolytic multivitamins and neomycin could improve the antioxidant capacity of transported lambs. In addition, the levels of the antioxidant enzymes and MDA of the EG could recover to the levels before transportation. As expected, there was an increase in the levels of SOD and T-AOC in sera, SOD and T-AOC in the jejunum, and GSH-Px, SOD, and T-AOC in the colon in the CG on day 14 compared to those during transportation, but they did not recover to the levels before transportation. The result showed that road transport still affected the antioxidant capacity of lambs on day 14. Electrolytic multivitamins and neomycin could potentially improve the antioxidant capacity of lambs, and the effect of electrolytic multivitamins was better; the possible reason was that the vitamins and electrolyte ions in electrolytic multivitamins had a synergistic effect, which potentially enhanced the activity of the antioxidant enzymes and improved the antioxidant capacity.

The Nrf2/Keap1/HO-1 pathway plays a crucial role in maintaining the redox balance. It is important to resist oxidative stress. Under normal conditions, Nrf2 and Keap1 inhibit each other mutually; when the body experiences oxidative stress, Nrf2 dissociates from Keap1, which interacts with antioxidant response elements, activates downstream related antioxidant factor HO-1, and promotes the antioxidant effects of the antioxidant enzyme [24,25,26]. In this experiment, by road transport, the mRNA expressions of Nrf2 in the jejunum and colon decreased significantly, while the mRNA expression of Keap1 increased. It showed that the activation of the Nrf2 pathway in the jejunum and colon was inhibited after lambs were transported, which led to oxidative stress in the jejunum and colon of lambs. Other research found that vitamins could induce activation of the Nrf2 antioxidant pathway, which also prevented oxidative stress and inflammation [27,28]. In this experiment, by treatment of electrolytic multivitamins, the mRNA expressions of Nrf2 in the jejunum and colon increased, while Keap1 in the jejunum and colon decreased; it showed that electrolytic multivitamins were beneficial for activating the Nrf2 signaling pathway and inhibiting the oxidative stress potentially. In addition, it found that the expressions of Nrf2, HO-1, and Keap1 in the EG could recover to the levels before transportation. However, on day 14, the mRNA expressions of Nrf2 and HO-1 in the jejunum and colon of the CG did not recover to the levels before transportation, which indicated that the oxidative stress caused by road transport in the jejunum and colon of lambs still existed on day 14; electrolytic multivitamins had a better effect on potentially reducing antioxidant system damage in transported lambs, which could protect the normal physiological function of the intestines.

Immunoglobulin participates in humoral immunity, and it is often used to evaluate the strength of immune function, which is one of the important indicators that reflects the level of disease resistance. sIgA is the main immunoglobulin in the intestines. It interacts with intestinal microorganisms and enhances the immune and barrier functions of intestinal mucosa to maintain homeostasis of the body [29]. Research showed that the levels of IgA, IgG, and IgM in serum reduced significantly after cows experienced heat stress [30,31]. Weaning stress led to a decrease in the content of sIgA in the intestines of piglets, which reduced the ability to resist pathogenic microorganisms [32]. The results indicated that stress affected the immunity of animals. In this experiment, by road transport, the concentrations of IgA, IgG, and sIgA in the jejunum and colon of lambs decreased significantly. It indicated that the intestinal immune function of lambs was inhibited by road transport, and the mucosal barrier function of the intestines may be disrupted. The concentration of IgG decreased significantly after transportation, which may be related to the HPA axis, which was activated by road transport. A significant decrease was found in the concentrations of IgA and sIgA. It may be due to road transport causing damage to the intestinal mucosal barrier, which affects the metabolism and absorption of nutrients in the intestine, thereby inhibiting the secretion of IgA and sIgA. Other research showed that vitamins could participate in humoral immunity and promote immunoglobulin synthesis [33,34]; adding electrolytes to feed could effectively alleviate heat stress in cows [35]. In this experiment, the concentrations of IgA and IgG in the jejunum increased significantly after treatment with electrolytic multivitamins. This study also found that the concentration of IgA in the jejunum and sIgA in the jejunum and colon of the CG on day 14 did not recover to the levels before transportation. It showed that electrolytic multivitamins had a better effect on potentially enhancing the mucosal immune function of the jejunum and colon and promoting humoral immunity of lambs. It may be caused by electrolytes and vitamins in electrolytic multivitamins. sIgA is the first line of defense against external pathogens, which can eliminate pathogens through non-specific immunity [36]. The concentration of sIgA in the jejunum and colon increased significantly after adding electrolytic multivitamins and neomycin; the intestinal mucosal immunity of lambs improved potentially, it could prevent pathogenic bacteria from adhering to the intestinal tract to cause intestinal infection and reduce the incidence rate of lambs. 

Under stress, the intestine is prone to becoming a target organ. Endotoxins, symbiotic bacteria, and pathogenic bacteria can enter the bloodstream through tight epithelial connections. It promotes the secretion of inflammatory factors and causes an imbalance between the pro-inflammatory and anti-inflammatory systems [37]. The abnormal expression of inflammatory factors is related to the immune function of the body. Large amounts of IL-1β could damage tissues and reduce the immunity of animals. IL-2 is produced by antigen-stimulated lymphocytes, which reflects the immune function of animals indirectly. IL-12 is a determinant of Th1 cellular immune response, which promotes the production of IFN- γ to exert antiviral effects. Wang et al. [38] found that the levels of IL-2 and IL-12 in rats increased significantly after they experienced heat stress for 7 days at 32 ℃, heat stress-induced inflammatory response in rats. In this experiment, the mRNA expressions of IL-1β, IL-2, and IL-12 in the jejunum and colon of lambs increased significantly after transportation. It indicated that road transport caused an imbalance in the inflammatory systems of the lambs. The possible reason was that lambs were influenced by stress factors; the body increased the secretion of IL-1β, IL-2, and IL-12 to participate in immune responses and mediate inflammatory responses by upregulating their expressions, it could potentially enhance immune function and alleviate damage caused by road transport [39]. Other research showed that vitamins could regulate the inflammatory response by controlling the release of anti-inflammatory and pro-inflammatory factors, thereby reducing the level of inflammatory factors [40]. In this experiment, by treatments of electrolytic multivitamin and neomycin, the mRNA expressions of IL-2 and IL-12 in the jejunum decreased significantly; electrolytic multivitamin also inhibited the mRNA expressions of IL-1 β in the jejunum and IL-2 in the colon. With the extension of time, the mRNA expressions of IL-1β, IL-2, and IL-12 in the jejunum and colon of the EG recovered firstly to the levels before transportation. However, the mRNA expressions of IL-1β, IL-2, and IL-12 in the jejunum of the CG on day 14 did not recover to the levels before transportation. It indicated that electrolytic multivitamins and neomycin could effectively downregulate the expressions of IL-2 and IL-12 in the jejunum and colon, electrolytic multivitamins had a better effect on alleviating inflammatory response potentially, there still was serious inflammatory reaction in the jejunum of lambs possibly on day 14 if no treatments to lambs, it may be related to electrolytic multivitamin could potentially activate the Nrf2/Keap1/HO-1 signaling pathway.

Apoptosis is a necessary physiological phenomenon for maintaining tissue development and function in the body. However, cell apoptosis happens excessively and could cause serious damage to tissues and organs [41]. Histiocytes will experience cell disintegration and fragmentation when histiocyte occurs apoptosis; the collapsed cells exist in tissues and organs with the form of apoptotic bodies, which can induce Caspase-3 activation, trigger cascade reaction, and initiate the process of apoptosis [42]. Bcl-2 can exert anti-apoptotic effects through various pathways, such as inhibiting cytochrome C and apoptosis-inducing factors [43]. Activated Bax can antagonize Bcl-2 and prevent Bcl-2 from exerting anti-apoptotic effects [44]. Wei et al. [45] found that the protein expressions of Bax and Caspase 3 increased, while the expression of Bcl-2 decreased after mice experienced heat stress. In this experiment, by road transport, the mRNA expressions of Bax in the jejunum and colon, as well as the mRNA expression of Caspase3 in the jejunum of lambs increased significantly, while the mRNA expression of Bcl-2 decreased significantly. The results showed that road transport promoted the abnormal apoptosis process of the intestine, which aggravated the overall degree of damage to the intestine. Other research showed that vitamins could effectively promote the expression of Bax and inhibit the expressions of Caspase3 and Bcl-2 in cells, which slowed apoptosis [46,47]. In this experiment, electrolytic multivitamins inhibited the mRNA expressions of Bax and Caspase3 in the jejunum, and colon promoted the mRNA expression of Bcl-2 in the jejunum. In addition, the mRNA expressions of Bax, Bcl-2, and Caspase3 in the jejunum and colon of the EG recovered firstly to the levels before transportation. On day 14, the mRNA expressions of Bax and Bcl-2 in the jejunum of the CG did not recover to the levels before transportation. The results showed that electrolytic multivitamins had a better effect on enhancing the anti-apoptotic ability of cells. It could enhance intestinal resistance to damage and help maintain normal intestinal function.

## 5. Conclusions

Road transport led to a decrease in the antioxidant capacity and immunoglobulin levels of lambs, while the levels of inflammatory factors and apoptosis in the jejunum and colon of lambs increased. It also inhibited the activation of the Nrf2 pathway in the jejunum and colon and even exacerbated damage to the jejunum and colon. On day 14, the levels of the antioxidant capacity, immunity, and apoptosis of lambs did not recover to the levels before transportation. Electrolytic multivitamins had a better effect on potentially improving the antioxidant level of lambs and inhibiting the release of inflammatory factors. It also potentially reduced the expression of apoptotic factors and alleviated damage in the jejunum and colon. 

## Figures and Tables

**Table 1 animals-14-00824-t001:** Ingredients and nutrient composition of the basal diet (dry matter basis).

Ingredients	Content %	Nutrient Composition	Content
Cracked corn	55.00	Neutral Detergent Fibers (%)	33.33
Soybean meal	20.00	Crude Protein (%)	18.06
Peanut seeding	12.50	Acidic Detergent Fibers (%)	14.49
Peanut meal	9.00	Metabolizable Energy (MJ/kg) ^2^	12.50
Premix ^1^	2.50	Ca (%)	0.75
NaCl	0.50	*p* (%)	0.31
Baking soda	0.50		

^1^ Per kg premix contained the following: vitamin A 15,356 IU, vitamin D 4300 IU, vitamin E 50 mg, Fe 88.70 mg, Zn 70.90 mg, Mn 51.80 mg, Cu 13.75 mg, Se 0.23 mg, I 1.50 mg, and Co 0.49 mg. ^2^ Metabolizable energy was a calculated value, while the others were measured values.

**Table 2 animals-14-00824-t002:** The main components and contents of electrolytic multivitamins.

Components	Contents
Vitamin C	≥2000 IU/kg
Vitamin B2	≥750 mg/kg
Vitamin A	30,000–5,000,000 IU/kg
Vitamin D3	75,000–20,000,000 IU/kg
Vitamin E	≥500 IU/kg
Vitamin B1	≥500 mg/kg
Water	≤10%
Folate	30 mg/kg
Taurine	20,000 mg/kg
Zn	1000 mg/kg
Mn	1000 mg/kg
Fe	1000 mg/kg
Cu	600 mg/kg

**Table 3 animals-14-00824-t003:** Gene primer information.

Primer	Sequence (5′→3′)		Length (bp)	Annealing Temperature (°C)	Accession Number
IL-1β	F	ACAGATGAAGAGCTGCACCC	161	58	443,539
	R	AGACATGTTCGTAGGCACGG			
IL-2	F	GTTGCAAACGGTGCACCTAC	122	58	443,401
	R	GAGAGCTTGAGGTTCTCGGG			
IL-12	F	GCTTCGCAGCCTCCTCC	136	59	443,472
	R	CCTCAGCAGGTTTTGGGAGT			
CAT	F	TTGCGGGCCATCTGAAAG	101	53	100,307,035
	R	AAGAGCCTGGATGCGGGAG			
SOD	F	GGTAAACACGGTTTCCAT	261	53	692,639
	R	CAAGTCATCAGGGTCAGC			
Nrf2	F	AAGGTCCTCCCCATCCATGA	193	58	443,276
	R	GCTCAACCCAGACTGGAGAC			
Keap1	F	CGTGGAGACAGAAACGTGGA	159	59	101,113,845
	R	CAGGTGTCTGTGTCTGGGTC			
HO-1	F	CGATGGGTCCTCACACTCAG	75	59	101,120,910
	R	GCACACTCGCATTCACATGG			
Bax	F	TGCCAGCAAACTGGTGCTCAA	243	60	443,059
	R	GCACTCCAGCCACAAAGATGGT			
Bcl-2	F	CGCTGAAGCGAAGCTGTAGA	92	60	101,119,602
	R	CGTTGAGCCTGAAAGCTGTTT			
Caspase3	F	AATGCAAGAAGCAGGGCACCCA	155	60	443,031
	R	GGGTTACAGCGATGCAGAAGGTTCA			
β-actin	F	CAGTCGGTTGGATCGAGCAT	151	59	443,340
	R	AGAAGGAGGGTGGCTTTTGG			

**Table 4 animals-14-00824-t004:** Effect of different treatments on the antioxidant capacity in sera of transported lambs.

Items	Groups	Time					*p*-Values		
		−2d	0d	7d	14d	Total	Time	Treatment	Interaction
SOD (U/mL)	CG	204.20 ± 10.77 ^a^	142.58 ± 6.97 ^Bb^	159.40 ± 10.27 ^b^	169.65 ± 9.46 ^b^	168.96 ± 6.75	<0.010	0.017	0.501
	EG	196.81 ± 13.63	175.45 ± 8.53 ^A^	186.04 ± 10.37	198.41 ± 10.22	189.18 ± 5.41			
	NG	197.00 ± 10.70 ^a^	153.35 ± 7.54 ^ABb^	161.68 ± 10.89 ^b^	181.49 ± 9.62 ^ab^	173.38 ± 5.95			
	Total	199.34 ± 6.36	157.13 ± 5.51	169.04 ± 6.48	183.18 ± 6.10				
MDA (nmol/mL)	CG	6.64 ± 0.38 ^b^	10.20 ± 1.08 ^a^	8.79 ± 0.70 ^ab^	8.13 ± 0.41 ^Aab^	8.44 ± 0.44	<0.010	0.019	0.695
	EG	6.79 ± 0.53	8.14 ± 0.69	7.35 ± 0.73	6.48 ± 0.35 ^B^	7.19 ± 0.31			
	NG	6.84 ± 0.63 ^b^	9.53 ± 0.53 ^a^	8.15 ± 0.58 ^ab^	7.44 ± 0.29 ^ABb^	7.99 ± 0.33			
	Total	6.76 ± 0.28	9.29 ± 0.49	8.10 ± 0.39	7.35 ± 0.26				
GSH-Px (U/mL)	CG	131.84 ± 8.30	108.50 ± 5.78	117.82 ± 6.08	123.99 ± 6.04	120.54 ± 3.62	0.069	0.165	0.925
	EG	130.60 ± 7.18	124.05 ± 8.96	129.97 ± 6.76	133.40 ± 6.87	129.51 ± 3.53			
	NG	126.86 ± 7.37	115.95 ± 5.46	122.81 ± 3.86	127.79 ± 6.37	123.36 ± 2.91			
	Total	129.77 ± 4.12	116.17 ± 4.07	123.53 ± 3.33	128.39 ± 3.59				
CAT (U/mL)	CG	0.77 ± 0.03	0.58 ± 0.11	0.70 ± 0.24	0.74 ± 0.08	0.70 ± 0.06	0.159	0.266	0.918
	EG	0.75 ± 0.02	0.72 ± 0.18	0.95 ± 0.33	1.08 ± 0.11	0.87 ± 0.12			
	NG	0.70 ± 0.05	0.48 ± 0.16	0.65 ± 0.12	0.96 ± 0.25	0.70 ± 0.07			
	Total	0.74 ± 0.02	0.59 ± 0.09	0.77 ± 0.14	0.92 ± 0.11				
T-AOC (U/mL)	CG	0.29 ± 0.02 ^a^	0.17 ± 0.01 ^Bc^	0.21 ± 0.01 ^bc^	0.23 ± 0.02 ^Bb^	0.23 ± 0.01	<0.010	<0.010	0.347
	EG	0.28 ± 0.02	0.23 ± 0.01 ^A^	0.25 ± 0.03	0.30 ± 0.04 ^A^	0.27 ± 0.01			
	NG	0.28 ± 0.02 ^a^	0.21 ± 0.01 ^Ab^	0.22 ± 0.01 ^b^	0.25 ± 0.03 ^Bab^	0.24 ± 0.01			
	Total	0.28 ± 0.01	0.21 ± 0.00	0.23 ± 0.01	0.26 ± 0.01				

Note: ^abc^ Different lowercase letters superscripted in the same row indicate significant differences between different treatments (*p* < 0.05). ^AB^ The superscript values in the same column with different uppercase letters indicate significant differences at different times (*p* < 0.05). The same letter or no letter indicates no significant difference (*p* > 0.05). There is no meaning between uppercase and lowercase letters.

**Table 5 animals-14-00824-t005:** Effect of different treatments on the jejunal and colonic antioxidant capacity of transported lambs.

Items		Groups	Time					*p*-Values		
			−2d	0d	7d	14d	Total	Time	Treatment	Interaction
Jejunum	GSH-Px (U/g)	CG	4272.54 ± 424.49 ^a^	2677.12 ± 212.66 ^b^	3124.78 ± 316.62 ^b^	3304.09 ± 284.91 ^ab^	3344.63 ± 222.09	<0.010	<0.440	0.968
		EG	4322.18 ± 429.24	2794.13 ± 284.62	3526.44 ± 315.13	3955.46 ± 482.23	3649.55 ± 237.75			
		NG	4499.46 ± 359.46 ^a^	2932.41 ± 212.41 ^b^	3285.34 ± 310.11 ^b^	3583.22 ± 390.33 ^ab^	3575.11 ± 223.47			
		Total	4364.72 ± 205.73	2801.22 ± 125.06	3312.19 ± 167.48	3614.26 ± 218.49				
	SOD (U/g)	CG	1202.02 ± 132.21 ^a^	630.72 ± 90.16 ^b^	752.42 ± 41.34 ^Bb^	811.85 ± 69.81 ^b^	849.25 ± 74.98	<0.010	0.017	0.883
		EG	1276.40 ± 105.72 ^a^	840.96 ± 68.96 ^b^	979.44 ± 59.99 ^Ab^	1025.74 ± 88.25 ^ab^	1030.39 ± 59.00			
		NG	1340.50 ± 105.97 ^a^	719.87 ± 61.30 ^b^	827.62 ± 46.85 ^ABb^	915.56 ± 67.26 ^b^	950.89 ± 77.61			
		Total	1272.64 ± 61.02	730.52 ± 48.11	853.16 ± 41.71	917.72 ± 48.84				
	MDA (nmol/g)	CG	46.76 ± 2.83 ^c^	81.85 ± 3.68 ^Aa^	70.22 ± 3.26 ^Ab^	61.74 ± 4.20 ^b^	65.14 ± 4.14	<0.010	<0.010	0.247
		EG	45.25 ± 1.01 ^b^	61.49 ± 2.34 ^Ba^	54.52 ± 3.58 ^Bab^	46.70 ± 3.80 ^b^	57.21 ± 3.46			
		NG	43.80 ± 1.83 ^c^	71.10 ± 4.87 ^ABa^	61.50 ± 2.92 ^ABab^	52.44 ± 4.61 ^bc^	51.99 ± 2.32			
		Total	45.27 ± 1.10	71.48 ± 3.49	62.08 ± 2.80	53.62 ± 3.04				
	T-AOC (μmol/mL)	CG	0.20 ± 0.01 ^a^	0.14 ± 0.01 ^b^	0.15 ± 0.01 ^b^	0.16 ± 0.01 ^Bb^	0.16 ± 0.01	<0.010	0.019	0.939
		EG	0.21 ± 0.01	0.16 ± 0.02	0.17 ± 0.03	0.19 ± 0.01 ^A^	0.18 ± 0.01			
		NG	0.21 ± 0.01 ^a^	0.17 ± 0.01 ^b^	0.17 ± 0.01 ^ab^	0.20 ± 0.01 ^Aab^	0.19 ± 0.01			
		Total	0.21 ± 0.00	0.15 ± 0.01	0.16 ± 0.01	0.18 ± 0.01				
Colon	GSH-Px (U/g)	CG	4132.23 ± 297.75 ^a^	2827.76 ± 193.04 ^b^	3084.56 ± 193.57 ^b^	3289.40 ± 216.07 ^b^	3333.49 ± 176.96	<0.010	0.075	0.933
		EG	4176.81 ± 377.81	3375.81 ± 204.78	3552.28 ± 312.19	3963.06 ± 320.70	3766.99 ± 163.45			
		NG	4165.16 ± 351.27 ^a^	3008.08 ± 211.41 ^b^	3195.69 ± 146.68 ^b^	3470.47 ± 200.25 ^ab^	3459 ± 85 ± 167.25			
		Total	4158.07 ± 172.08	3070.55 ± 129.72	3277.51 ± 134.21	3574.31 ± 161.03				
	SOD (U/g)	CG	1165.63 ± 83.32 ^a^	623.04 ± 30.36 ^Bb^	694.00 ± 69.47 ^Bb^	774.66 ± 43.42 ^Bb^	814.33 ± 68.29	<0.010	<0.010	0.157
		EG	1335.44 ± 64.47^a^	1012.40 ± 62.18 ^Ab^	1021.90 ± 59.88 ^Ab^	1282.88 ± 90.73 ^Aa^	1014.72 ± 57.87			
		NG	1194.94 ± 89.52 ^a^	774.05 ± 42.83 ^Bc^	946.69 ± 78.70 ^Abc^	1143.19 ± 43.21 ^Aab^	1163.16 ± 53.59			
		Total	1232.01 ± 47.74	803.16 ± 61.34	887.53 ± 60.63	1066.91 ± 82.12				
	MDA (nmol/g)	CG	39.04 ± 3.93 ^b^	61.31 ± 3.71 ^Aa^	49.20 ± 4.28 ^ab^	44.33 ± 3.94 ^b^	48.47 ± 3.01	<0.010	0.042	0.494
		EG	38.52 ± 4.43	45.31 ± 3.34 ^B^	41.73 ± 3.69	38.67 ± 4.06	41.06 ± 1.86			
		NG	38.28 ± 4.50 ^b^	60.08 ± 4.06 ^Aa^	45.12 ± 4.21 ^b^	41.03 ± 3.48 ^b^	46.13 ± 3.08			
		Total	38.61 ± 2.15	55.57 ± 3.17	45.35 ± 2.30	41.34 ± 2.09				
	T-AOC (μmol/mL)	CG	0.20 ± 0.01 ^a^	0.15 ± 0.01 ^b^	0.14 ± 0.01 ^b^	0.14 ± 0.01 ^b^	0.16 ± 0.01	<0.010	0.537	0.995
		EG	0.20 ± 0.02	0.16 ± 0.02	0.16 ± 0.02	0.15 ± 0.01	0.17 ± 0.01			
		NG	0.20 ± 0.01 ^a^	0.15 ± 0.01 ^b^	0.15 ± 0.01 ^b^	0.14 ± 0.01 ^b^	0.16 ± 0.01			
		Total	0.20 ± 0.01	0.15 ± 0.01	0.15 ± 0.01	0.14 ± 0.01				

Note: ^abc^ Different lowercase letters superscripted in the same row indicate significant differences between different treatments (*p* < 0.05). ^AB^ The superscript values in the same column with different uppercase letters indicate significant differences at different times (*p* < 0.05). The same letter or no letter indicates no significant difference (*p* > 0.05). There is no meaning between uppercase and lowercase letters.

**Table 6 animals-14-00824-t006:** Effects of different treatments on the mRNA expressions of antioxidant factors in the jejunum and colon of transported lambs.

Items		Groups	Time					*p*-Value		
			−2d	0d	7d	14d	Total	Time	Treatment	Interaction
Jejunum	CAT	CG	1.00 ± 0.00 ^a^	0.65 ± 0.05 ^b^	0.70 ± 0.05 ^b^	0.75 ± 0.07 ^b^	0.78 ± 0.04	<0.010	0.091	0.915
		EG	1.00 ± 0.00 ^a^	0.75 ± 0.06 ^b^	0.80 ± 0.07 ^b^	0.90 ± 0.07 ^ab^	0.86 ± 0.03			
		NG	1.00 ± 0.00 ^a^	0.69 ± 0.08 ^b^	0.74 ± 0.07 ^b^	0.79 ± 0.06 ^b^	0.80 ± 0.04			
		Total	1.00 ± 0.00	0.69 ± 0.04	0.75 ± 0.04	0.81 ± 0.04				
	SOD	CG	1.00 ± 0.00 ^a^	0.54 ± 0.08 ^b^	0.56 ± 0.07 ^Bb^	0.70 ± 0.05 ^Bb^	0.70 ± 0.05	<0.01	<0.01	0.123
		EG	1.00 ± 0.00 ^a^	0.72 ± 0.09 ^b^	0.89 ± 0.05 ^Aab^	1.10 ± 0.09 ^Aa^	0.93 ± 0.04			
		NG	1.00 ± 0.00 ^a^	0.58 ± 0.09 ^b^	0.69 ± 0.08 ^ABb^	0.91 ± 0.09 ^ABa^	0.79 ± 0.05			
		Total	1.00 ± 0.00	0.61 ± 0.05	0.71 ± 0.05	0.90 ± 0.06				
Colon	CAT	CG	1.00 ± 0.00 ^a^	0.67 ± 0.07 ^b^	0.70 ± 0.06 ^b^	0.74 ± 0.07 ^b^	0.78 ± 0.04	<0.010	0.076	0.893
		EG	1.00 ± 0.00 ^a^	0.76 ± 0.06 ^b^	0.81 ± 0.05 ^b^	0.89 ± 0.05 ^ab^	0.86 ± 0.03			
		NG	1.00 ± 0.00 ^a^	0.71 ± 0.08 ^b^	0.75 ± 0.03 ^b^	0.79 ± 0.06 ^b^	0.81 ± 0.03			
		Total	1.00 ± 0.00	0.71 ± 0.04	0.75 ± 0.03	0.81 ± 0.04				
	SOD	CG	1.00 ± 0.00 ^a^	0.66 ± 0.08 ^b^	0.74 ± 0.07 ^b^	0.76 ± 0.04 ^Bb^	0.79 ± 0.04	<0.01	0.011	0.736
		EG	1.00 ± 0.00	0.82 ± 0.10	0.88 ± 0.08	0.96 ± 0.05 ^A^	0.92 ± 0.03			
		NG	1.00 ± 0.00 ^a^	0.73 ± 0.05 ^b^	0.79 ± 0.06 ^b^	0.86 ± 0.05 ^ABb^	0.85 ± 0.03			
		Total	1.00 ± 0.00	0.74 ± 0.04	0.80 ± 0.04	0.86 ± 0.03				

Note: ^ab^ Different lowercase letters superscripted in the same row indicate significant differences between different treatments (*p* < 0.05). ^AB^ The superscript values in the same column with different uppercase letters indicate significant differences at different times ( *p* < 0.05). The same letter or no letter indicates no significant difference (*p* > 0.05). There is no meaning between uppercase and lowercase letters.

**Table 7 animals-14-00824-t007:** Effects of different treatments on mRNA expressions of key factors in the Nrf2 pathway in the jejunum and colon of transported lambs.

Items		Groups	Time					*p*-Values		
			−2d	0d	7d	14d	Total	Time	Treatment	Interaction
Jejunum	Nrf2	CG	1.00 ± 0.00 ^a^	0.51 ± 0.03 ^Bc^	0.61 ± 0.08 ^c^	0.82 ± 0.06 ^Bb^	0.73 ± 0.04	<0.010	<0.010	0.144
		EG	1.00 ± 0.00 ^a^	0.72 ± 0.04 ^Ab^	0.81 ± 0.09 ^b^	1.10 ± 0.05 ^Aa^	0.91 ± 0.04			
		NG	1.00 ± 0.00 ^a^	0.55 ± 0.03 ^Bc^	0.71 ± 0.06 ^b^	0.95 ± 0.06 ^ABa^	0.80 ± 0.04			
		Total	1.00 ± 0.00	0.59 ± 0.03	0.71 ± 0.04	0.96 ± 0.04				
	HO-1	CG	1.00 ± 0.00 ^a^	0.57 ± 0.04 ^c^	0.67 ± 0.04 ^c^	0.80 ± 0.06 ^b^	0.76 ± 0.04	<0.010	0.073	0.849
		EG	1.00 ± 0.00 ^a^	0.64 ± 0.04 ^b^	0.75 ± 0.06 ^b^	0.92 ± 0.05 ^a^	0.83 ± 0.04			
		NG	1.00 ± 0.00 ^a^	0.61 ± 0.03 ^c^	0.69 ± 0.03 ^c^	0.82 ± 0.07 ^b^	0.78 ± 0.04			
		Total	1.00 ± 0.00	0.61 ± 0.02	0.71 ± 0.03	0.85 ± 0.04				
	Keap1	CG	1.00 ± 0.00 ^b^	1.36 ± 0.06 ^Aa^	1.20 ± 0.10 ^ab^	1.07 ± 0.07 ^b^	1.16 ± 0.04	<0.01	0.040	0.753
		EG	1.00 ± 0.00	1.13 ± 0.05 ^B^	1.09 ± 0.07	0.96 ± 0.06	1.05 ± 0.03			
		NG	1.00 ± 0.00 ^b^	1.24 ± 0.05 ^ABa^	1.15 ± 0.07 ^ab^	1.03 ± 0.08 ^b^	1.11 ± 0.03			
		Total	1.00 ± 0.00	1.25 ± 0.04	1.15 ± 0.05	1.02 ± 0.04				
Colon	Nrf2	CG	1.00 ± 0.00 ^a^	0.63 ± 0.04 ^b^	0.65 ± 0.04 ^b^	0.70 ± 0.04 ^Bb^	0.75 ± 0.04	<0.010	<0.010	0.179
		EG	1.00 ± 0.00 ^a^	0.76 ± 0.05 ^b^	0.81 ± 0.08 ^b^	0.99 ± 0.07 ^Aa^	0.89 ± 0.03			
		NG	1.00 ± 0.00 ^a^	0.65 ± 0.05 ^c^	0.72 ± 0.05 ^bc^	0.85 ± 0.08 ^ABab^	0.80 ± 0.04			
		Total	1.00 ± 0.00	0.68 ± 0.03	0.73 ± 0.03	0.85 ± 0.04				
	HO-1	CG	1.00 ± 0.00 ^a^	0.60 ± 0.06 ^c^	0.70 ± 0.06 ^bc^	0.83 ± 0.05 ^b^	0.78 ± 0.04	<0.010	0.255	0.977
		EG	1.00 ± 0.00 ^a^	0.68 ± 0.07 ^c^	0.80 ± 0.05 ^bc^	0.90 ± 0.05 ^ab^	0.84 ± 0.03			
		NG	1.00 ± 0.00 ^a^	0.64 ± 0.05 ^b^	0.74 ± 0.07 ^b^	0.89 ± 0.06 ^b^	0.82 ± 0.04			
		Total	1.00 ± 0.00	0.64 ± 0.03	0.74 ± 0.03	0.87 ± 0.03				
	Keap1	CG	1.00 ± 0.00 ^c^	1.59 ± 0.13 ^a^	1.31 ± 0.04 ^Ab^	1.21 ± 0.09 ^bc^	1.28 ± 0.06	<0.010	0.011	0.741
		EG	1.00 ± 0.00 ^b^	1.37 ± 0.12 ^a^	1.09 ± 0.05 ^Bb^	1.03 ± 0.05 ^b^	1.12 ± 0.04			
		NG	1.00 ± 0.00 ^c^	1.44 ± 0.09 ^a^	1.21 ± 0.06 ^ABb^	1.08 ± 0.07 ^bc^	1.18 ± 0.05			
		Total	1.00 ± 0.00	1.47 ± 0.07	1.20 ± 0.04	1.11 ± 0.04				

Note: ^abc^ Different lowercase letters superscripted in the same row indicate significant differences between different treatments (*p* < 0.05). ^AB^ The superscript values in the same column with different uppercase letters indicate significant differences at different times (*p* < 0.05). The same letter or no letter indicates no significant difference (*p* > 0.05). There is no meaning between uppercase and lowercase letters.

**Table 8 animals-14-00824-t008:** Effects of different treatments on intestinal immunoglobulin in transported lambs.

Items		Groups	Time					*p*-Values		
			−2d	0d	7d	14d	Total	Time	Treatment	Interaction
Jejunum	IgA (mg/g)	CG	2.43 ± 0.22 ^a^	1.37 ± 0.17 ^b^	1.49 ± 0.11 ^b^	1.71 ± 0.13 ^Bb^	1750.19 ± 142.58	<0.010	0.011	0.567
		EG	2.40 ± 0.24	1.91 ± 0.20	2.15 ± 0.25	2.41 ± 0.16 ^A^	2217.40 ± 110.16			
		NG	2.35 ± 0.25 ^a^	1.34 ± 0.23 ^b^	1.78 ± 0.20 ^ab^	2.00 ± 0.13 ^ABab^	1866.17 ± 140.95			
		Total	2393.11 ± 119.24	1539.97 ± 136.78	1805.70 ± 136.18	2039.57 ± 122.67				
	IgG (mg/g)	CG	703.33 ± 52.35 ^a^	489.48 ± 27.65 ^c^	538.86 ± 33.17 ^Bbc^	631.33 ± 36.03 ^ab^	590.75 ± 29.80	<0.010	0.035	0.613
		EG	697.38 ± 55.10	554.22 ± 35.75	705.96 ± 34.92 ^A^	736.46 ± 44.44	673.50 ± 97.47			
		NG	709.14 ± 45.48 ^a^	521.16 ± 38.85 ^b^	636.52 ± 33.27 ^ABab^	681.68 ± 56.48^a^	637.12 ± 99.60			
		Total	703.28 ± 25.63	521.62 ± 19.58	627.11 ± 29.54	683.16 ± 27.83				
	IgM (mg/g)	CG	18.70 ± 3.5	10.45 ± 2.52	12.40 ± 2.40	14.76 ± 3.31	14.08 ± 1.57	0.129	0.075	0.903
		EG	18.48 ± 3.26	12.61 ± 1.95	14.98 ± 2.28	17.25 ± 3.81	19.07 ± 1.42			
		NG	18.53 ± 3.41	16.62 ± 2.67	18.93 ± 3.07	22.18 ± 2.93	15.83 ± 1.42			
		Total	18.57 ± 1.70	13.23 ± 1.50	15.44 ± 4.84	18.07 ± 2.01				
	sIgA (μg/g)	CG	342.43 ± 21.75 ^a^	167.20 ± 8.63 ^Bc^	167.88 ± 14.77 ^Bc^	290.37 ± 14.18 ^Bb^	214.97 ± 24.05	<0.010	<0.010	0.558
		EG	358.40 ± 28.81 ^a^	258.26 ± 16.80 ^Ab^	330.23 ± 26.01 ^Aab^	378.44 ± 29.20 ^Aa^	331.33 ± 17.56			
		NG	350.60 ± 23.22^a^	197.13 ± 18.47 ^Bc^	267.51 ± 15.26 ^Ab^	300.48 ± 20.62 ^Aab^	278.93 ± 18.77			
		Total	350.47 ± 12.60	207.53 ± 15.42	255.21 ± 25.55	323.10 ± 17.80				
	IgA (mg/g)	CG	2.43 ± 0.19 ^a^	1.52 ± 0.13 ^b^	1.67 ± 0.17 ^b^	1.94 ± 0.18 ^ab^	1.89 ± 0.13	<0.010	0.116	0.853
		EG	2.35 ± 0.21	1.88 ± 0.21	2.15 ± 0.22	2.42 ± 0.26	2.20 ± 0.11			
		NG	2.31 ± 0.25 ^a^	1.50 ± 0.10 ^b^	1.76 ± 0.10 ^ab^	2.10 ± 0.17 ^a^	1.92 ± 0.12			
		Total	2.36 ± 0.11	1.63 ± 0.10	1.86 ± 0.11	2.15 ± 0.13				
Colon	IgG (mg/g)	CG	747.80 ± 88.62 ^a^	510.64 ± 40.44 ^b^	552.42 ± 32.93 ^b^	665.01 ± 39.02 ^ab^	618.97 ± 36.64	<0.010	0.551	0.996
		EG	743.44 ± 71.48	560.51 ± 44.91	606.16 ± 46.22	732.72 ± 53.97	660.71 ± 33.51			
		NG	747.18 ± 70.10 ^a^	527.66 ± 38.62 ^b^	581.19 ± 38.43 ^ab^	706.58 ± 45.80 ^a^	640.65 ± 34.36			
		Total	746.14 ± 38.60	532.94 ± 21.96	579.92 ± 21.25	701.44 ± 25.33				
	IgM(mg/g)	CG	17.69 ± 2.87	12.29 ± 3.17	14.81 ± 2.46	16.42 ± 1.97	15.30 ± 1.29	0.366	0.174	0.972
		EG	17.70 ± 3.22	13.81 ± 2.83	16.02 ± 2.33	18.36 ± 2.59	19.07 ± 1.34			
		NG	17.84 ± 2.54	17.83 ± 2.58	19.54 ± 3.03	21.08 ± 3.72	16.47 ± 1.29			
		Total	17.74 ± 1.45	14.64 ± 1.66	16.79 ± 1.49	18.62 ± 1.58				
	sIgA (μg/g)	CG	355.95 ± 21.74 ^a^	188.01 ± 18.03 ^Bc^	233.59 ± 18.733 ^Bbc^	293.41 ± 17.34 ^Bb^	267.74 ± 20.70	<0.010	<0.010	0.138
		EG	388.19 ± 33.88	320.89 ± 21.49 ^A^	395.63 ± 26.92 ^A^	454.04 ± 29.97 ^A^	389.69 ± 18.55			
		NG	365.47 ± 27.14 ^a^	233.09 ± 16.86 ^Bc^	302.30 ± 19.00 ^Bbc^	337.17 ± 24.88 ^Bb^	309.51 ± 17.70			
		Total	369.87 ± 14.81	247.33 ± 21.67	310.51 ± 25.90	361.54 ± 26.77				

Note: ^abc^ Different lowercase letters superscripted in the same row indicate significant differences between different treatments (*p* < 0.05). ^AB^ The superscript values in the same column with different uppercase letters indicate significant differences at different times (*p* < 0.05). The same letter or no letter indicates no significant difference (*p* > 0.05). There is no meaning between uppercase and lowercase letters.

**Table 9 animals-14-00824-t009:** Effects of different treatments on mRNA expression of inflammatory factors in the jejunum and colon of transported lambs.

Items		Groups	Time					*p*-Values		
			−2d	0d	7d	14d	Total	Time	Treatment	Interaction
Jejunum	IL-1β	CG	1.00 ± 0.00 ^b^	1.30 ± 0.08 ^a^	1.26 ± 0.09 ^a^	1.19 ± 0.03 ^Aa^	1.27 ± 0.07	<0.010	0.030	0.678
		EG	1.00 ± 0.00 ^b^	1.22 ± 0.09 ^a^	1.13 ± 0.04 ^ab^	0.98 ± 0.06 ^Bb^	1.19 ± 0.07			
		NG	1.00 ± 0.00 ^c^	1.27 ± 0.07 ^a^	1.22 ± 0.08 ^ab^	1.12 ± 0.03 ^Abc^	1.20 ± 0.06			
		Total	1.00 ± 0.00	1.46 ± 0.09	1.28 ± 0.07	1.15 ± 0.06				
	IL-2	CG	1.00 ± 0.00 ^b^	1.58 ± 0.18 ^a^	1.41 ± 0.04 ^a^	1.36 ± 0.04 ^Aa^	1.34 ± 0.06	<0.010	0.013	0.612
		EG	1.00 ± 0.00 ^bc^	1.37 ± 0.08 ^a^	1.26 ± 0.09 ^ab^	1.07 ± 0.08 ^Bbc^	1.18 ± 0.05			
		NG	1.00 ± 0.00 ^b^	1.43 ± 0.06 ^a^	1.36 ± 0.07 ^a^	1.15 ± 0.04 ^Bb^	1.24 ± 0.04			
		Total	1.00 ± 0.00	1.46 ± 0.07	1.34 ± 0.04	1.19 ± 0.04				
	IL-12	CG	1.00 ± 0.00 ^c^	1.57 ± 0.07 ^a^	1.49 ± 0.08 ^a^	1.25 ± 0.07 ^Ab^	1.33 ± 0.05	<0.010	<0.010	0.286
		EG	1.00 ± 0.00 ^b^	1.38 ± 0.11 ^a^	1.25 ± 0.05 ^a^	0.87 ± 0.07 ^Bb^	1.13 ± 0.05			
		NG	1.00 ± 0.00 ^b^	1.41 ± 0.10 ^a^	1.37 ± 0.10 ^a^	1.01 ± 0.09 ^Bb^	1.20 ± 0.06			
		Total	1.00 ± 0.00	1.45 ± 0.05	1.37 ± 0.05	1.05 ± 0.06				
Colon	IL-1β	CG	1.00 ± 0.00 ^b^	1.47 ± 0.16 ^a^	1.21 ± 0.12 ^ab^	1.08 ± 0.11 ^ab^	1.19 ± 0.04	<0.010	0.531	0.957
		EG	1.00 ± 0.00	1.44 ± 0.16	1.39 ± 0.13	1.26 ± 0.15	1.08 ± 0.03			
		NG	1.00 ± 0.00 ^b^	1.46 ± 0.15 ^a^	1.23 ± 0.11 ^ab^	1.12 ± 0.05 ^ab^	1.15 ± 0.03			
		Total	1.00 ± 0.00	1.27 ± 0.04	1.20 ± 0.04	1.10 ± 0.03				
	IL-2	CG	1.00 ± 0.00 ^c^	1.50 ± 0.05 ^Aa^	1.40 ± 0.07 ^ab^	1.19 ± 0.12 ^bc^	1.27 ± 0.05	<0.010	0.023	0.449
		EG	1.00 ± 0.00	1.23 ± 0.03 ^B^	1.05 ± 0.16	1.06 ± 0.05	1.09 ± 0.04			
		NG	1.00 ± 0.00 ^b^	1.46 ± 0.08 ^Aa^	1.16 ± 0.21 ^ab^	1.03 ± 0.03 ^b^	1.16 ± 0.07			
		Total	1.00 ± 0.00	1.40 ± 0.04	1.20 ± 0.09	1.09 ± 0.05				
	IL-12	CG	1.00 ± 0.00 ^b^	1.44 ± 0.11^a^	1.28 ± 0.12 ^b^	1.14 ± 0.09 ^ab^	1.22 ± 0.06	<0.010	0.169	0.960
		EG	1.00 ± 0.00	1.25 ± 0.11	1.11 ± 0.12	1.03 ± 0.07	1.10 ± 0.05			
		NG	1.00 ± 0.00 ^b^	1.34 ± 0.10 ^a^	1.19 ± 0.11 ^ab^	1.07 ± 0.10 ^ab^	1.15 ± 0.05			
		Total	1.00 ± 0.00	1.35 ± 0.06	1.19 ± 0.07	1.08 ± 0.05				

Note: ^abc^ Different lowercase letters superscripted in the same row indicate significant differences between different treatments (*p* < 0.05). ^AB^ The superscript values in the same column with different uppercase letters indicate significant differences at different times (*p* < 0.05). The same letter or no letter indicates no significant difference (*p* > 0.05). There is no meaning between uppercase and lowercase letters.

**Table 10 animals-14-00824-t010:** The effect of different treatments on apoptosis factors in the jejunum and colon of transported lambs.

Items		Groups	Time					*p*-Values		
			−2d	0d	7d	14d	Total	Time	Treatment	Interaction
Jejunum	Bax	CG	1.00 ± 0.00 ^c^	1.56 ± 0.06 ^Aa^	1.36 ± 0.06 ^Ab^	1.25 ± 0.04 ^b^	1.29 ± 0.05	<0.010	<0.010	0.275
		EG	1.00 ± 0.00 ^b^	1.25 ± 0.05 ^Ba^	1.10 ± 0.05 ^Bab^	1.02 ± 0.08 ^b^	1.09 ± 0.03			
		NG	1.00 ± 0.00 ^c^	1.41 ± 0.10 ^ABa^	1.26 ± 0.05 ^ABab^	1.11 ± 0.11 ^bc^	1.20 ± 0.05			
		Total	1.00 ± 0.00	1.41 ± 0.05	1.24 ± 0.04	1.13 ± 0.05				
	Bcl-2	CG	1.00 ± 0.00 ^a^	0.66 ± 0.03 ^c^	0.72 ± 0.04 ^c^	0.89 ± 0.04 ^Bb^	0.82 ± 0.03	<0.010	0.012	0.360
		EG	1.00 ± 0.00 ^a^	0.72 ± 0.04 ^b^	0.80 ± 0.05 ^b^	1.08 ± 0.06 ^Aa^	0.90 ± 0.04			
		NG	1.00 ± 0.00 ^a^	0.70 ± 0.04 ^b^	0.77 ± 0.05 ^b^	0.98 ± 0.04 ^ABa^	0.86 ± 0.03			
		Total	1.00 ± 0.00	0.69 ± 0.02	0.76 ± 0.03	0.98 ± 0.03				
	Caspase3	CG	1.00 ± 0.00 ^c^	1.57 ± 0.07 ^Aa^	1.32 ± 0.14 ^ab^	1.20 ± 0.13 ^bc^	1.27 ± 0.06	<0.010	0.010	0.646
		EG	1.00 ± 0.00 ^c^	1.26 ± 0.03 ^Ba^	1.14 ± 0.07 ^b^	1.00 ± 0.04 ^c^	1.10 ± 0.03			
		NG	1.00 ± 0.00 ^c^	1.46 ± 0.05 ^Aa^	1.23 ± 0.11 ^b^	1.14 ± 0.09 ^bc^	1.21 ± 0.05			
		Total	1.00 ± 0.00	1.43 ± 0.04	1.23 ± 0.06	1.11 ± 0.05				
Colon	Bax	CG	1.00 ± 0.00 ^c^	1.65 ± 0.07 ^Aa^	1.35 ± 0.06 ^Ab^	1.16 ± 0.15 ^bc^	1.29 ± 0.06	<0.010	<0.010	0.231
		EG	1.00 ± 0.00 ^bc^	1.32 ± 0.06 ^Ba^	1.08 ± 0.07 ^Bb^	0.88 ± 0.03 ^c^	1.07 ± 0.04			
		NG	1.00 ± 0.00 ^c^	1.50 ± 0.07 ^ABa^	1.19 ± 0.05 ^ABb^	0.98 ± 0.05 ^c^	1.17 ± 0.05			
		Total	1.00 ± 0.00	1.49 ± 0.05	1.21 ± 0.04	1.00 ± 0.04				
	Bcl-2	CG	1.00 ± 0.00 ^a^	0.76 ± 0.05 ^b^	0.84 ± 0.08 ^ab^	0.92 ± 0.06 ^ab^	0.88 ± 0.03	<0.010	0.250	0.981
		EG	1.00 ± 0.00 ^a^	0.82 ± 0.05 ^b^	0.93 ± 0.05 ^ab^	1.02 ± 0.07 ^a^	0.94 ± 0.03			
		NG	1.00 ± 0.00 ^a^	0.80 ± 0.03 ^b^	0.89 ± 0.06 ^ab^	0.98 ± 0.07 ^a^	0.92 ± 0.03			
		Total	1.00 ± 0.00	0.80 ± 0.02	0.89 ± 0.03	0.98 ± 0.04				
	Caspase3	CG	1.00 ± 0.00 ^b^	1.33 ± 0.13 ^a^	1.26 ± 0.06 ^Aa^	1.15 ± 0.08 ^ab^	1.19 ± 0.05	<0.010	0.026	0.730
		EG	1.00 ± 0.00	1.15 ± 0.07	1.05 ± 0.06 ^B^	1.00 ± 0.06	1.05 ± 0.03			
		NG	1.00 ± 0.00 ^b^	1.31 ± 0.07 ^a^	1.12 ± 0.07 ^ABab^	1.09 ± 0.09 ^b^	1.13 ± 0.04			
		Total	1.00 ± 0.00	1.26 ± 0.06	1.14 ± 0.04	1.08 ± 0.04				

Note: ^abc^ Different lowercase letters superscripted in the same row indicate significant differences between different treatments (*p* < 0.05). ^AB^ The superscript values in the same column with different uppercase letters indicate significant differences at different times (*p* < 0.05). The same letter or no letter indicates no significant difference (*p* > 0.05). There is no meaning between uppercase and lowercase letters.

## Data Availability

The data that support the findings of this study are available from the corresponding author upon reasonable request at any time.

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
