# Peer review of "The Effects of Electrolytic Multivitamins and Neomycin on Antioxidant Capacity and Intestinal Damage in Transported Lambs"

_animals, 2024, doi:10.3390/ani14060824_

Round 1

Reviewer 1 Report (Previous Reviewer 1)

Comments and Suggestions for Authors

Dear Authors,

I appreciate the efforts made by the authors in improving the revised submission titled "The Effects of Electrolytic Multivitamin and Neomycin on Antioxidant Capacity and Intestinal Damage in Transported Lambs."

The revised version of the article has shown improvement in several aspects. However, there are still several grammar and syntax errors throughout the manuscript which I believe the editorial team can manage. Additionally, the experimental design still appears somewhat disjointed, although the authors have provided justification for administering multivitamins two days before animal transportation. It would be beneficial for the authors to provide references supporting this decision.

Authors have made revisions to the results, but there is inconsistency in presentation, for example, the inclusion of SEM only in Table 2. I suggest adopting the same presentation format in all tables for consistency.

Specific comments and suggestions are provided below:

80-82 Please provide necessary references from previous literature. Additionally, consider starting both supplementations on the same day.

83 Replace "Feed" with "fed".

Table 1 Replace "ingredient level" with "nutrient composition".

Table 1 The revised NaCl values cannot be the same as in the inclusion list. Authors need to be careful with their revisions. If available, authors should mention Na and Cl separately in the composition section.

159-160 The statement regarding the levels of SOD and T-AOC in CG and NG needs revision for clarity. Consider specifying the timing or treatment effects.

Table 2 I suggest removing individual errors with means if pooled error is provided. Additionally, consider adding overall means of treatments regardless of time points for easier interpretation. Improve consistency in presenting SEM across all tables.

302 Replace "Researches" with "earlier research".

307 Replace "SOD" with "The SOD".

319 Replace "Other papers" with "other research" or "earlier work".

323-324 Adapt the style of this sentence to the result and conclusion sections, and include findings related to Neomycin treatment.

326 Add that as expected, in the CG group, there was an increase in ------ during transportation stress. Introduce this information earlier in the discussion.

Conclusion Authors must align their conclusions with their results.

Author Response

Reviewer 2 Report (Previous Reviewer 2)

Comments and Suggestions for Authors

Dear Authors,

Your manuscript is now suitable for publication at Animals in present form.

Yours sincerely,

Reviewer.

Author Response

Dear reviewer,

We are honored to receive your recognition of our work. We are appreciated for your warm work earnestly. Once again, thank you very much for your comments and recognition.

Yours sincerely,

Cui Xia

This manuscript is a resubmission of an earlier submission. The following is a list of the peer review reports and author responses from that submission.

Round 1

Reviewer 1 Report

Comments and Suggestions for Authors

Dear Authors,

I appreciate the effort you put into your manuscript titled "The effects of electrolytic multivitamin and neomycin on anti-2 oxidant capacity and intestinal damage in transported lambs." I would like to offer constructive feedback to enhance the quality of your work.

Your article sheds light on stress mitigation through supplementation in lambs; however, there are notable issues with English language proficiency throughout the document. The study design, in particular, lacks clarity. For instance, you presented NG means in the table at -2 days, but the design statement suggests no Neomycin supplementation, posing a significant limitation to the study.

Furthermore, there are repetitions in the presentation of methods, and the results in the text section lack clarity. The data analysis techniques and presentation of results need improvement. Specific comments and suggestions are provided below:

Comments/Suggestions:

Line No.49: Revise "it is harmful to animals" for clarity.

Line No. 50-52: Merge sentences for better understanding: "which becomes a target organ for stress easily. The stability of intestinal function is crucial for the body."

Line No.60: Clarify the intention behind "less papers about." Do authors mean there is limited published literature available on ----?

Line No. 61-62: Revise or delete "and the prevention and treatment of transportation stress becomes a hot topic" for clarity.

Line No.65-68: Rephrase statements to avoid contradiction.

Line No.70-71: Rephrase for clarity: "on them and the possible mechanism by establishing a transport stress model."

Line No.78: Replace "•" with "/".

Line No.: Replace with "were."

Line No.82: Provide a detailed explanation for not starting neomycin 2 days before transportation to maintain consistency in the experimental design.

Table 1:

Replace "nutrient level" with "ingredient level."

Delete "100" at the end of the inclusion column.

Mention units for NDF, Crude Protein, Acidic Detergent Fibers, Ca, and Phosphorus.

Replace NaCl with Na+ and Cl separately along with units in the table.

In the footnotes, write the full names of vitamins for improved readability.

Replace "Metabolic Energy" with "Metabolizable Energy" in the table and footnotes.

Line No.10-112:

"Blood Sample Collection" is already mentioned at Line 84-86. Delete either one to reduce redundancy.

Line No. 113-115:Avoid repetition.

Line No. 148-15:Improve data presentation by including treatment x time interaction.

Present overall means of each treatment, followed by individual time points.

Table 4: Instead of individual error of the means, include pooled SEM.

Consider including NG means for SOD.

Address the discrepancy regarding NG means at -2 days and the lack of Neomycin supplementation in the design statement.

Line No. 158-159: Improve the presentation of results for better reader comprehension.

I hope these suggestions contribute to the enhancement of your manuscript. 

Reviewer 2 Report

Comments and Suggestions for Authors

Dear Authors, 

The manuscript titled 'The effects of electrolytic multivitamin and neomycin on antioxidant capacity and intestinal damage in transported lambs' is aiming to evaluate the effects of road transport on the antioxidant capacity and intestinal damage of lambs by exploring whether electrolytic multivitamin and neomycin had effect on them. Its main contributions are related to provide new ideas for improving antioxidant capacity and reducing intestinal damage of transport lambs under stress conditions. 

The article has several areas of weakness, specially due to the lack of relevant information in Introduction, inaccuracies in Material and Methods, missing controls in Results, editing in Conclusions and incorrectness of References. Major reviewed is needed to justify the relevance of the topic under research and to clarify the experimental design applied (protocol number approval is lacking and days of sampling are confusing). Tables need also to be checked to get uniformity in decimals and Conclusions rewritten. Finally, References need to be reviewed according to Animals' Guidelines for authors.

Please, see below a list of specific comments/suggestions to be addressed.

L28 Replace '(P<0.01)' by '(P<0.01) under a CG diet'. 

L29 Replace 'Caspase3' by ' Caspase3' levels and 'MDA increased' by 'MDA, however, increased'.

L30 Replace 'colon decreased' by 'colon also decreased'.

L33 Replace 'Bcl2' by 'Bcl2, however,'.

L35 Replace 'decreased' by 'also decreased'.

L38 Replace 'Road' by 'In summary, road', 'can cause antioxidant' by 'can cause a decrease in antioxidant' and 'decreasing, and' by 'while'.

L39 Replace 'oxidative damage increasing' by 'an increase in oxidative damage'.

L60 Replace 'less papers' by 'low number of papers are searching'.

L62 Add references to justify that it's a hot topic.

L65 Replace 'Less research' by 'Lack of research is observed in the literature'.

L66 Add references to justify this statement.

L68 Add references to justify this statement.

L78 Replace 'd . lamb' by  'd per lamb'.

L81 Replace 'd . lamb' by  'd per lamb'.

L77-L83 Explain why did you choose different days for supplying with EG and NG,

L87-L89 Explain how many days animals wait until they were anesthetized.

L84-L89 Delete sentences repeated in lines L110-L115.

L89 Add protocol number approval for collection of animals' samples.

L93 Replace 'mg, Co' by 'mg and Co'.

L103 Replace 'transportation,' by 'transportation was provided,'.

L104 Replace 'provided' by 'given'.

L105 Replace 'provided' by 'included'.

L114 Explain how did you get jejunal and colonic tissues.

L133 Replace 'Kit according' by 'Kit was done according'.

L163 Delete 'significantly'.

Tables 4-10. Add references to mean and standard error of the mean. Use 1 decimal for small number in average and 2 decimals for big number.

L178-L185 Delete 'significantly'.

L224-L228 Delete 'significantly'.

L271-L273 Delete 'significantly'.

L291-L296 Delete 'significantly'.

L322 Replace 'significantly, the level' by 'significantly, while the level'.

L353 Replace 'significantly,' by 'significantly, while'.

L373 Replace 'reduced significantly' by 'is significantly reduced'. 

L383 Replace 'may be due' by 'It may be due to'.

L430 Replace 'is necessary' by 'is a necessary'.

L433 Replace 'Collapsed' by 'collapsed'.

L456 Replace 'led to the antioxidant ... decerased' by 'is linked to a decrease in the antioxidant ...'.

L458 Replace 'increased' by 'is increased'.

L459 Replace 'exarcebated' by 'even exarcebated'.

L460 Replace 'they will could not recover' by 'lambs could be not recovered'. 

L461 Replace 'better' by 'a positive'.

L486-L584 Check all References according to Animals' guidelines for authors.

Yours sincerely,

Reviewer.